# CNGA3 acts as a cold sensor in hypothalamic neurons

Viktor V Feketa[1,2,3], Yury A Nikolaev[1], Dana K Merriman[4], Sviatoslav N Bagriantsev[1]*, Elena O Gracheva[1,2,3]*

[1]Department of Cellular and Molecular Physiology, Yale University School of Medicine, New Haven, United States; [2]Department of Neuroscience, Yale University School of Medicine, New Haven, United States; [3]Program in Cellular Neuroscience, Neurodegeneration and Repair, Yale University School of Medicine, New Haven, United States; [4]Department of Biology, University of Wisconsin-Oshkosh, Oshkosh, United States

**Abstract** Most mammals maintain their body temperature around 37°C, whereas in hibernators it can approach 0°C without triggering a thermogenic response. The remarkable plasticity of the thermoregulatory system allowed mammals to thrive in variable environmental conditions and occupy a wide range of geographical habitats, but the molecular basis of thermoregulation remains poorly understood. Here we leverage the thermoregulatory differences between mice and hibernating thirteen-lined ground squirrels (*Ictidomys tridecemlineatus*) to investigate the mechanism of cold sensitivity in the preoptic area (POA) of the hypothalamus, a critical thermoregulatory region. We report that, in comparison to squirrels, mice have a larger proportion of cold-sensitive neurons in the POA. We further show that mouse cold-sensitive neurons express the cyclic nucleotide-gated ion channel CNGA3, and that mouse, but not squirrel, CNGA3 is potentiated by cold. Our data reveal CNGA3 as a hypothalamic cold sensor and a molecular marker to interrogate the neuronal circuitry underlying thermoregulation.

**\*For correspondence:**
slav.bagriantsev@yale.edu (SNB);
elena.gracheva@yale.edu (EOG)

**Competing interests:** The authors declare that no competing interests exist.

## Introduction

The preoptic area of the hypothalamus (POA) is a key thermoregulatory region in the brain of vertebrates. It integrates signals from peripheral thermoreceptors and detects chemical cues produced by infectious agents to orchestrate physiological and behavioural thermoregulatory responses (*Angilletta et al., 2019*; *Madden and Morrison, 2019*; *Siemens and Kamm, 2018*; *Tan and Knight, 2018*). The POA of various species of mammals also contains neurons that respond to changes in local temperature (*Hardy et al., 1964*; *Hori et al., 1980a*; *Kelso et al., 1982*). Activation of POA neurons by cooling can occur in the absence of synaptic connections and in vitro, suggesting a cell-autonomous mechanism of cold sensitivity (*Abe et al., 2003*; *Hori et al., 1980b*). Although the physiological role of cold-sensing neurons remains obscure, they are thought to contribute, together with warm-sensing POA neurons, to the feedback thermoregulatory pathway by directly responding to changes in intracranial temperature (*Madden and Morrison, 2019*; *Siemens and Kamm, 2018*; *Tan and Knight, 2018*). Cold-sensitive POA neurons are found not only in mammals, but also in fish, reptiles and birds, suggesting an ancient origin and evolutionary conservation of this process (*Cabanac et al., 1967*; *Nelson and Prosser, 1981*; *Simon et al., 1977*). Understanding the function of cold-sensitive neurons is hampered by a lack of information about their molecular identity and mechanism of cold detection.

Here, we report that cold-activated mouse POA neurons are marked by the expression of the cyclic nucleotide-gated ion channel CNGA3. The channel is potentiated by cooling when heterologously expressed in various cell types, whereas pharmacological inhibition of CNGA3 suppresses

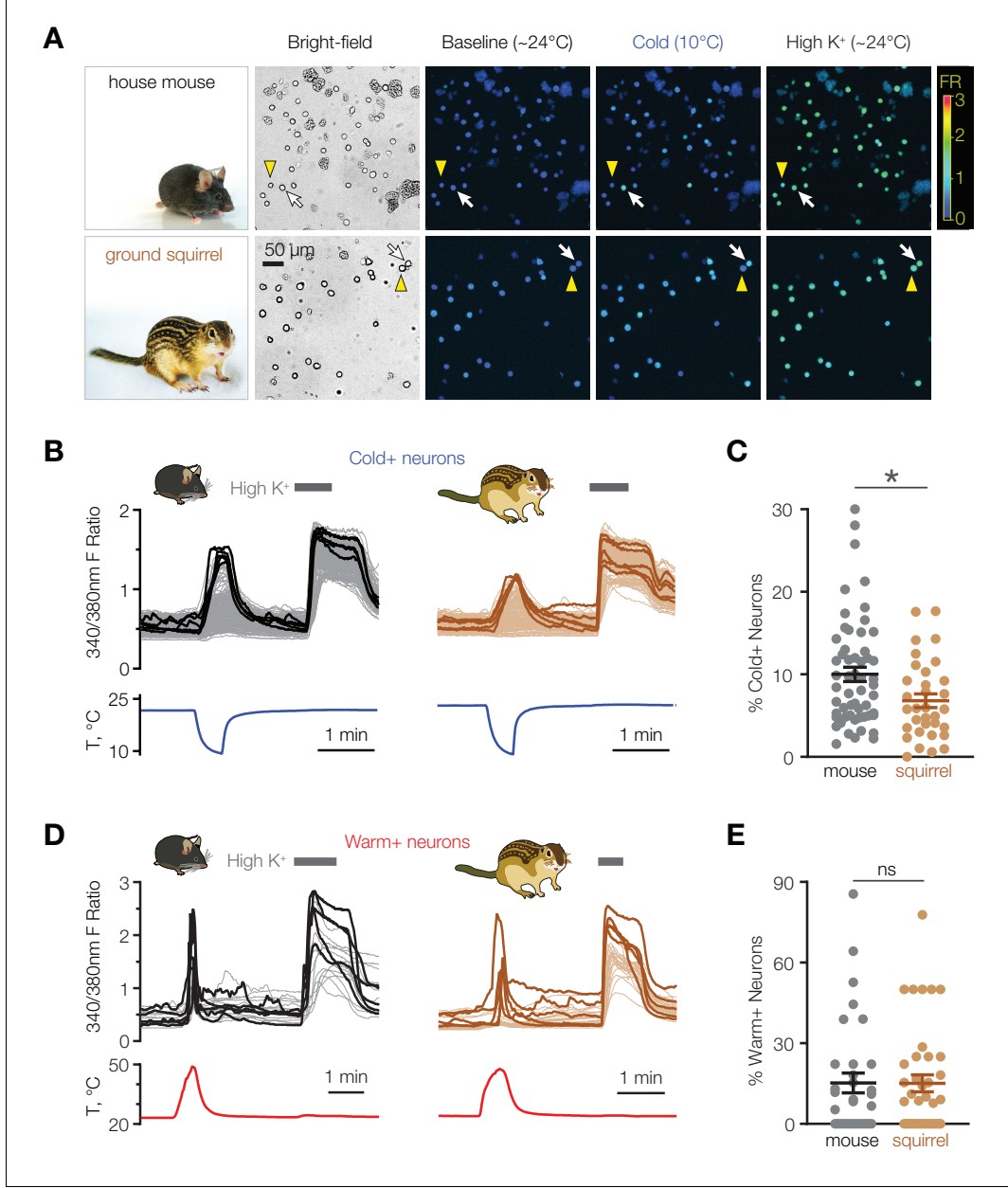

**Figure 1.** Mice have a higher proportion of cold-sensitive POA neurons than ground squirrels. (**A**) Representative bright-field and ratiometric calcium images of POA neurons from house mouse and thirteen-lined ground squirrel at baseline (~24°C), after cooling to 10°C, and after perfusion with 116 mM [K$^+$] ("High K$^+$") demonstrate cold-sensitive (white arrows) and cold-insensitive (yellow arrowheads) POA neurons. Ratiometric images are pseudocolored according to the fluorescence ratio (FR) scale bar. Photos courtesy of the Gracheva laboratory. (**B**) Fluorescent ratio traces in response to a cooling ramp for individual POA neurons from a representative coverslip from mouse (*left panel*, N = 182 neurons) and ground squirrel (*right panel*, N = 129 neurons). POA neurons were harvested and plated on multiple coverslips per animal as shown in *Figure 2A*. Five neurons with the highest cold response amplitude are highlighted with thick lines. (**C**) Percentage of cold-sensitive ("Cold+") POA neurons in mice and ground squirrels. *p<0.05, Mann-Whitney test. Each data point represents one coverslip. The horizontal line and error bars denote mean and SEM. N = 56 coverslips from 13 mice vs 33 coverslips from nine ground squirrels. (**D**) Fluorescent ratio traces in response to a warming ramp for individual POA neurons from a representative coverslip from mouse (*left panel*, N = 18 neurons) and ground squirrel (*right panel*, N = 28 neurons). Five neurons with the highest warm response amplitude are highlighted with thick lines. (**E**) Percentage of warm-sensitive ("Warm+") POA neurons in mice and ground squirrels. p>0.05 ('ns'), Mann-Whitney test. Each data point represents one coverslip. N = 33 coverslips from 6 mice vs 39 coverslips from 7 ground squirrels.

*Figure 1 continued on next page*

*Figure 1 continued*

The online version of this article includes the following source data for figure 1:

**Source data 1.** Percentage of cold-sensitive and warm-sensitive POA neurons in mice and ground squirrels.

cold sensitivity of POA neurons. In contrast, a CNGA3 orthologue from the POA of hibernating thirteen-lined ground squirrels is cold-insensitive, and squirrels have a smaller proportion of cold-sensitive neurons in the POA compared to mice. The correspondence of cold sensitivity of POA neurons and CNGA3 orthologues between mice and squirrels suggests that CNGA3 acts as a hypothalamic cold sensor with a potential role in thermoregulation.

## Results

### CNGA3 is enriched in cold-sensitive POA neurons in mice

We hypothesized that animal species that exhibit strong differences in their thermoregulatory physiology may give us clues to a possible role of cold-sensing neurons in the POA. To address this, we compared the cold sensitivity of POA neurons from mice with those from thirteen-lined ground squirrels, which in contrast to mice are able to drop their core body temperature close to 0℃ during torpor without mounting a thermogenic response (*Andrews, 2019*). Using ratiometric calcium imaging, we found that $10.0 \pm 0.9\%$ of dissociated mouse POA neurons responded to cooling (*Figure 1A–C*), consistent with previous studies (*Abe et al., 2003*). In contrast, active squirrels contained a significantly smaller population of cold-activated POA neurons ($6.8 \pm 0.8\%$, $p<0.05$ *vs.* mice, Mann-Whitney test, *Figure 1A–C*). The difference was specific to cold-sensing neurons because the proportion of warm-sensitive cells was similar in both mouse and squirrel POA (mice: $15.2 \pm 3.7\%$, squirrels: $15.1 \pm 3.1\%$, $p>0.05$, Mann-Whitney test, *Figure 1D and E*). These data suggest that a population of mouse POA neurons might possess a cold-sensing mechanism that is absent in squirrel POA.

To explore this possibility and examine the underlying mechanism, we looked for molecules that are preferentially expressed in cold-activated POA neurons in mice. Transcriptome analysis of pooled cold-sensitive versus cold-insensitive mouse POA neurons revealed that the most highly enriched transcript within the cold-sensitive neuronal population was the cyclic nucleotide-gated ion channel *Cnga3* (14-fold enrichment, *Figure 2A and B*). RNA in situ hybridization confirmed the expression of *Cnga3* in a subset of mouse POA neurons (*Figure 2C*). Together, these data reveal that mice contain a population of cold-sensitive POA neurons, which are enriched with CNGA3 channels.

We hypothesized that CNGA3 activity is necessary for cold sensitivity of the POA. This idea is supported by the findings that injection of the endogenous cyclic nucleotide-gated channel activator, cGMP, into the POA alters the firing rate and thermosensitivity of hypothalamic neurons and affects core body temperature in several species (*Wright et al., 2008*). Consistent with this, we found that l-*cis*-diltiazem, an inhibitor of CNGA3-containing heteromeric channels, drastically suppressed cold activation of mouse POA neurons (*Figure 3A and B*). At the same time, the inhibitor failed to affect calcium influx in response to depolarization by high extracellular potassium, demonstrating a specific effect on cold-sensing machinery (*Figure 3C*). Furthermore, the inhibitory effect of l-*cis*-diltiazem was specific to cold-sensing cells, as the drug failed to suppress activation of warm-sensitive cells (*Figure 3D–F*). These data strongly indicate that CNGA3 is necessary for cold sensitivity of mouse POA neurons.

### Cooling potentiates mouse, but not squirrel CNGA3

Several other non-selective cation channels are known to function as molecular temperature sensors (*Caterina et al., 1997*; *McKemy et al., 2002*; *Peier et al., 2002*; *Song et al., 2016*; *Tan and McNaughton, 2016*; *Togashi et al., 2006*), thus we suspected that CNGA3 could act as a cold sensor in the mouse POA. To test this, we cloned mouse *Cnga3* (mCNGA3) from the POA, expressed it in *Xenopus* oocytes, and measured its temperature sensitivity by two-electrode voltage clamp. Cooling the extracellular solution from 22℃ to 12℃ failed to stimulate mCNGA3, demonstrating that cold alone does not activate the channel (*Figure 4A*). However, cooling in the presence of a sub-

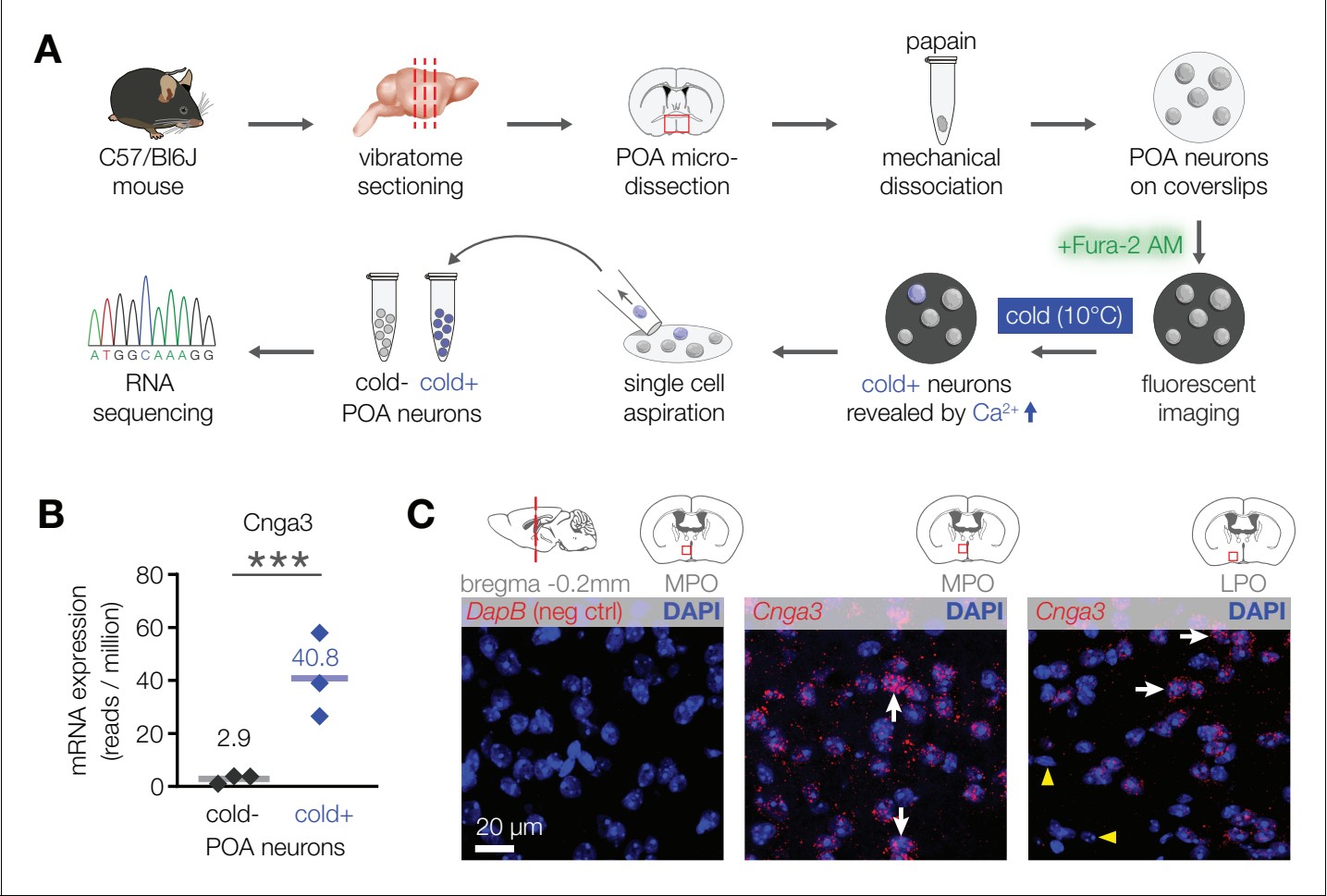

**Figure 2.** CNGA3 is enriched in cold-sensitive POA neurons in mice. (**A**) A schematic diagram of the imaging-guided collection and separation of cold-sensitive and cold-insensitive mouse POA neurons for differential transcriptomics. (**B**) Quantification of *Cnga3* transcript in cold-sensitive ("cold+") and cold-insensitive ("cold−") mouse POA neurons determined by RNA sequencing. ***$p<0.001$, GLM quasi-likelihood F-test (EdgeR). N = 3 independent biological replicates containing ~ 100–200 POA neurons each collected over 2–5 mice/independent neuron isolations. (**C**) RNA in situ hybridization images (maximal intensity projections of confocal Z-stacks) of the POA probed for *Cnga3* (*middle and right panels*) and DapB (negative control; *left panel*) reveal neurons with abundant *Cnga3* expression (white arrows) and neurons with no *Cnga3* expression (yellow arrowheads) in the medial preoptic area (MPO) and lateral preoptic area (LPO). Brain sections correspond to AP coordinate of bregma −0.2 mm. Images are representative of 8 fields-of-view from 4 sections from two independent procedures.

The online version of this article includes the following source data for figure 2:

**Source data 1.** Cnga3 expression in cold-sensitive and cold-insensitive mouse POA neurons.

threshold concentration (1 µM) of intracellular cGMP led to potent and reversible activation of mCNGA3 across a wide range of voltages (4-fold activation at 60 mV), consistent with the idea that cold facilitates cGMP-mediated activation of the channel (***Figure 4B and D***).

CNGA3 can co-assemble with CNGB1 or CNGB3 to form functional heteromeric channels with increased sensitivity to inhibition by l-*cis*-diltiazem compared to CNGA3 homomers (***Peng et al., 2004***; ***Peng et al., 2003***; ***Zhong et al., 2003***). The profound suppression of cold activation in the POA by l-*cis*-diltiazem suggests the presence of drug-sensitive heteromers. Our transcriptomic analysis revealed that *Cngb1*, but not *Cngb3*, is expressed in both cold-sensitive and insensitive POA neurons (***Figure 5A***). We thus co-expressed mouse CNGA3 and CNGB1 in oocytes and tested the effect of l-*cis*-diltiazem on cold-activated current in the presence of 1 µM of intracellular cGMP. Co-expression of mCNGA3 with mCNGB1 led to a twofold inhibition of cold-activated current by l-*cis*-diltiazem compared to mCNGA3 alone (***Figure 5B and C***). Since CNGB1 cannot form functional homomers, the drug-resistant fraction of the cold-activated current is likely due to the presence of

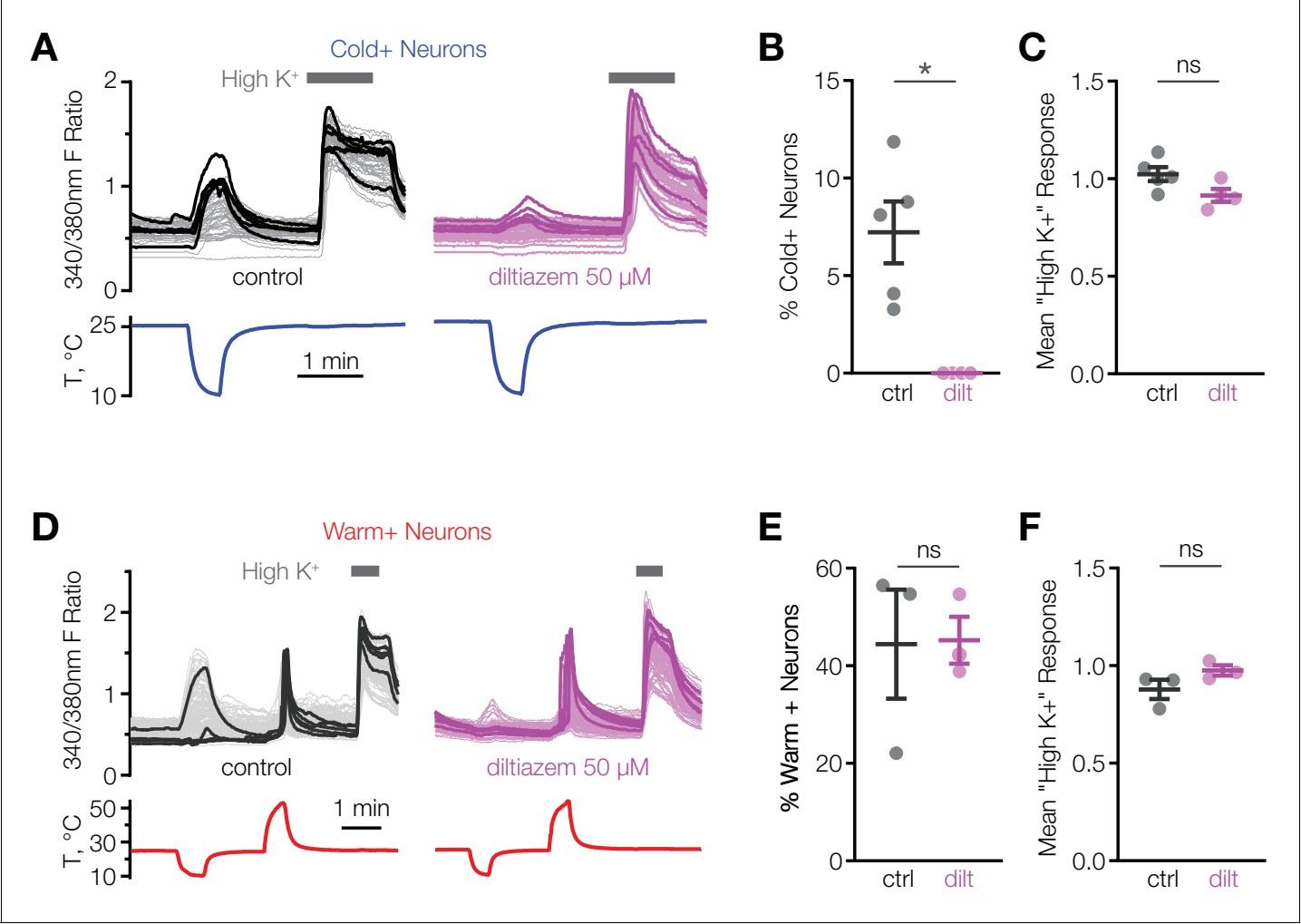

**Figure 3.** L-*cis*-diltiazem inhibits cold-activated neurons in mouse POA. (**A and D**) Fluorescent ratio traces in response to a cooling (**A**) or warming (**D**) ramp for individual POA neurons from a representative coverslip in the absence (*left panels*, A: N = 49 neurons, D: N = 145 neurons) or in the presence (right panels, A: N = 39 neurons, D: N = 103 neurons) of 50 μM l-*cis*-diltiazem. Five neurons with the highest cold or warm response amplitude are highlighted with thicker lines. (**B and E**) Percentage of cold-sensitive ("Cold+") or warm-sensitive ("Warm+") POA neurons in the absence ('ctrl') or in the presence of 50 μM l-*cis*-diltiazem ('dilt'). B: *p<0.05, E: p>0.05 ('ns'), Welch's t-test. Each data point represents one coverslip. The horizontal line and error bars denote mean and SEM. N = 4–5 coverslips from two mice in (**B**), 3 coverslips from one mouse in (**E**). (**C and F**) Per-coverslip average of the maximal response of all POA neurons recorded in the experiments in (**B**) and (**E**) to 116 mM [K$^+$] solution. p>0.05 ('ns'), Welch's t-test.
The online version of this article includes the following source data for figure 3:

**Source data 1.** Percentage of cold-sensitive and warm-sensitive POA neurons and their "High K" response amplitudes with and without l-cis-diltiazem.

mCNGA3 homomers (*Zhong et al., 2003*). Thus, CNGA3-CNGB1 heteromers can form a cold-activated l-*cis*-diltiazem-sensitive ion channel.

To test if cold activates other members of the cyclic nucleotide-gated channel family, we analysed CNGA2, a cGMP-activated ion channel (*Altenhofen et al., 1991*; *Gordon and Zagotta, 1995*; *Trudeau and Zagotta, 2003*), which we found to be expressed in cold-activated and cold-insensitive mouse POA neurons at similar levels (*Figure 6A*). In contrast to CNGA3, cold failed to activate mouse CNGA2 in oocytes in the presence of 1 μM intracellular cGMP (*Figure 6B and C*), while saturating 250 μM cGMP potently activated the channel (*Figure 6D*). This result demonstrates that cold activation is specific to CNGA3 among the cyclic nucleotide-gated channel family.

Having observed that mice have a larger proportion of cold-sensitive neurons in the POA than ground squirrels, we sought to test the cold sensitivity of the squirrel orthologue of CNGA3. Strikingly, cold failed to potentiate the activity of *Cnga3* cloned from squirrel POA (sqCNGA3, *Figure 4C*

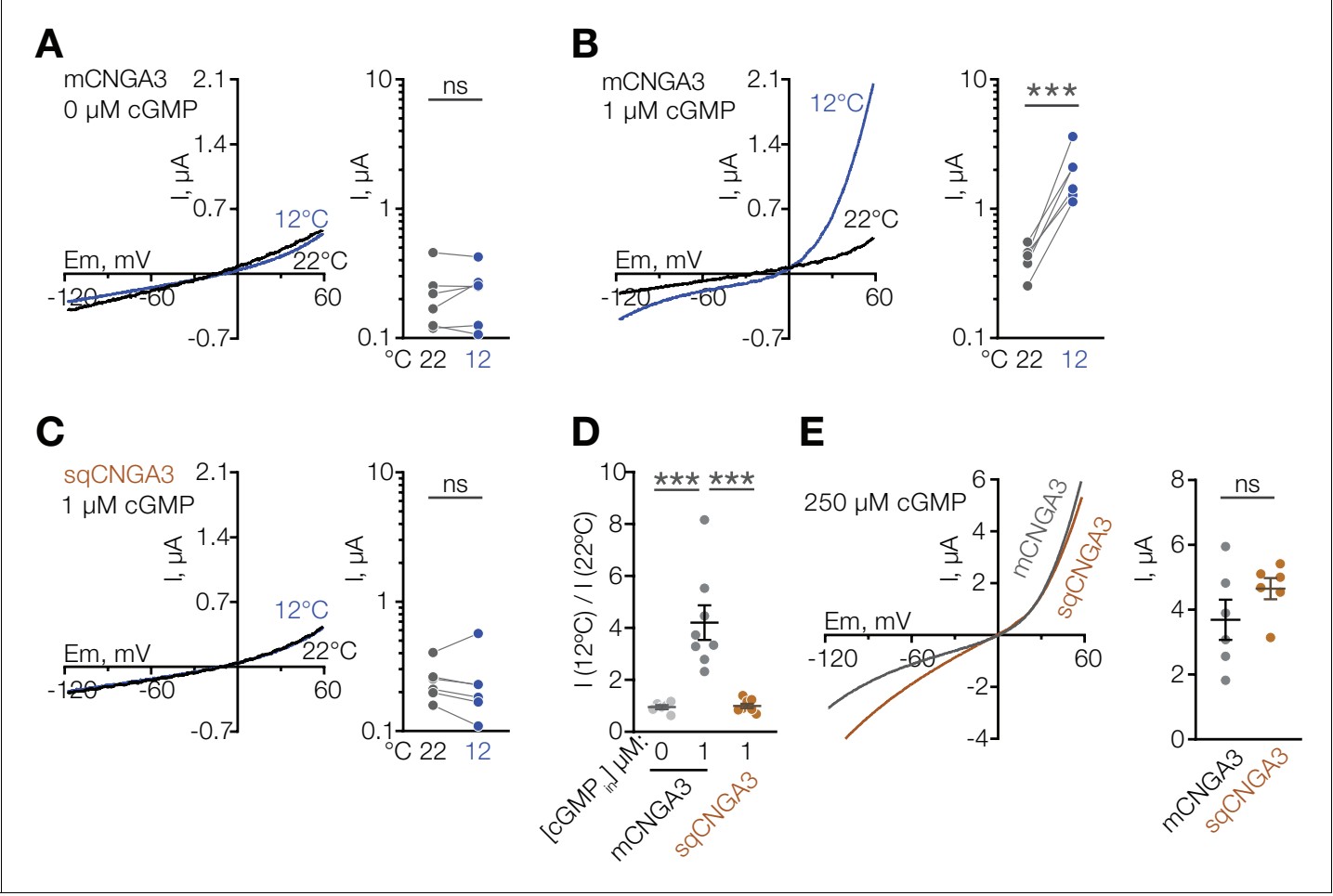

**Figure 4.** Cooling potentiates mouse but not squirrel CNGA3 in oocytes. (**A–C**) Exemplar whole-cell currents (*left*) and quantification of current amplitude at 60 mV (*right*) recorded in *Xenopus* oocytes expressing mouse or squirrel CNGA3 in response to a voltage ramp in the presence or absence of 1 μM intracellular cGMP. (**D**) Quantification of the magnitude of cold activation of CNGA3 current in oocytes at 60 mV in the presence of 1 μM intracellular cGMP. (**E**) Exemplar whole-cell current traces (*left*) and quantification of current amplitude at 60 mV (*right*) recorded from mouse or squirrel CNGA3 in response to a voltage ramp in the presence of saturating 250 μM intracellular cGMP. Data are mean ± SEM. NS, not significant, p>0.05; ***p<0.001, paired t-test (**A–C**), Dunnett's test (**D**), t-test (**F**). Each dot represents an individual cell.

The online version of this article includes the following source data for figure 4:

**Source data 1.** Cooling potentiates mouse but not squirrel CNGA3 in oocytes.

*and D*), even though activation with a saturating 250 μM cGMP revealed that sqCNGA3 was abundantly expressed on the surface of the oocytes (*Figure 4E*). To test if these observations were specific to frog oocytes, we expressed mouse and squirrel CNGA3 in HEK293T cells. In agreement with the data from oocytes, cooling from 22°C to 12°C reversibly potentiated mouse CNGA3 at positive and negative potentials in the presence of a sub-threshold 2 μM cGMP, but failed to affect squirrel CNGA3 (*Figure 7A–D* and *Figure 7—figure supplement 1*), even though both channels were abundantly expressed on the plasma membrane, as revealed by recordings in the presence of saturating 100 μM intracellular cGMP (*Figure 7E and F*).

Having established that cold specifically potentiates activity of mouse CNGA3 in various cell types, we performed a detailed characterization of its temperature dependence over the broad range of temperatures from 37°C to 10°C at a potential close to the physiological resting potential (−80 mV) in the presence of sub-threshold 1 μM intracellular cGMP. Cooling led to a robust activation of mCNGA3, with an apparent temperature threshold of 22.4 ± 0.8°C, and a 10-degree activation coefficient ($Q_{10}$) of 6.5 ± 0.5 (*Figure 8*). Together, our data demonstrate that cold specifically and reversibly potentiates the activity of mouse, but not squirrel CNGA3 in the presence of

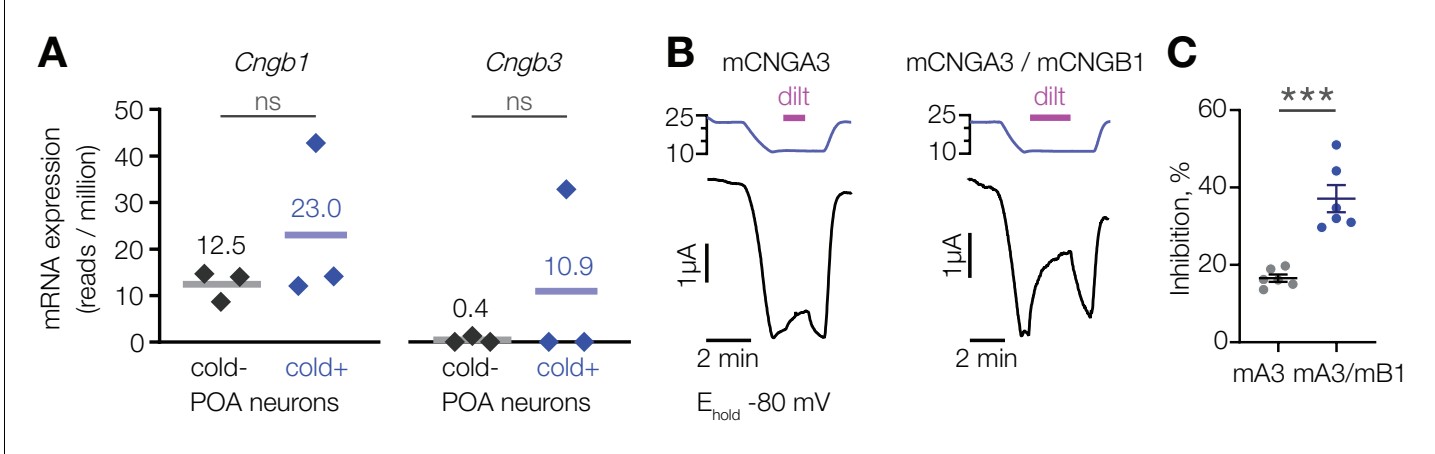

**Figure 5.** L-*cis*-diltiazem inhibits cold-activated current produced by CNGA3-CNGB1 heteromers. (**A**) Quantification of *Cngb1* and *Cngb3* transcripts in cold-sensitive ("cold+") and cold-insensitive ("cold−") mouse POA neurons determined by RNA sequencing. 'ns', not significant, p>0.05, GLM quasi-likelihood F-test (EdgeR). N = 3 independent biological replicates containing ~ 100–200 POA neurons each collected over 2–5 mice/independent neuron isolations. (**B**) Exemplar whole-cell currents obtained from *Xenopus* oocytes expressing mouse CNGA3 alone or with mouse CNGB1, recorded at −80 mV at different temperatures in the presence of 1 µM intracellular cGMP and 100 µM extracellular l-*cis*-diltiazem ('dilt'). (**C**) Quantification of maximum inhibition of cold-activated current by l-*cis*-diltiazem. Data are mean ± SEM. ***p<0.001, t-test. Each dot represents an individual cell. The online version of this article includes the following source data for figure 5:

**Source data 1.** L-*cis*-diltiazem inhibits cold-activated current produced by CNGA3-CNGB1 heteromers.

subthreshold intracellular cGMP, and that this effect is independent of cell type. Our results also agree with the notion that cold sensitivity of mouse POA neurons is mediated by heteromers and homomers of CNGA3.

Next, we aimed to clarify the mechanism of cold potentiation of mouse CNGA3, and hypothesized that cold decreases the effective concentration of cGMP required for channel opening. To test

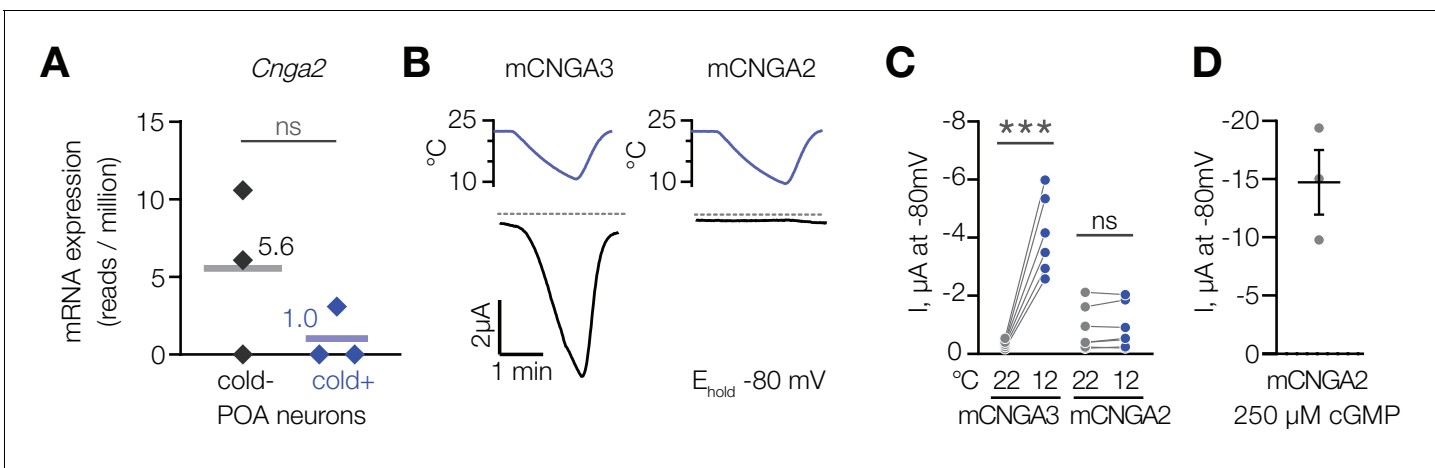

**Figure 6.** Cooling does not affect CNGA2 current. (**A**) Quantification of *Cnga2* transcript in cold-sensitive ("cold+") and cold-insensitive ("cold−") mouse POA neurons determined by RNA sequencing. 'ns', not significant, p>0.05, GLM quasi-likelihood F-test (EdgeR). N = 3 independent biological replicates containing ~ 100–200 POA neurons each collected over 2–5 mice/independent neuron isolations. (**B**) Exemplar whole-cell currents obtained from *Xenopus* oocytes expressing mouse CNGA3 and CNGA2, recorded at −80 mV at different temperatures in the presence of 1 µM intracellular cGMP. (**C**) Quantification of the effect of cooling on CNGA3 and CNGA2 current in the presence of 1 µM intracellular cGMP. (**D**) Quantification of mCNGA2 activity in the presence of 250 µM intracellular cGMP measured at −80 mV at 22°C. Data are mean ± SEM. 'ns', not significant, p>0.05; ***p<0.001, paired t-test. Each dot represents an individual cell.
The online version of this article includes the following source data for figure 6:

**Source data 1.** Cooling does not affect CNGA2 current.

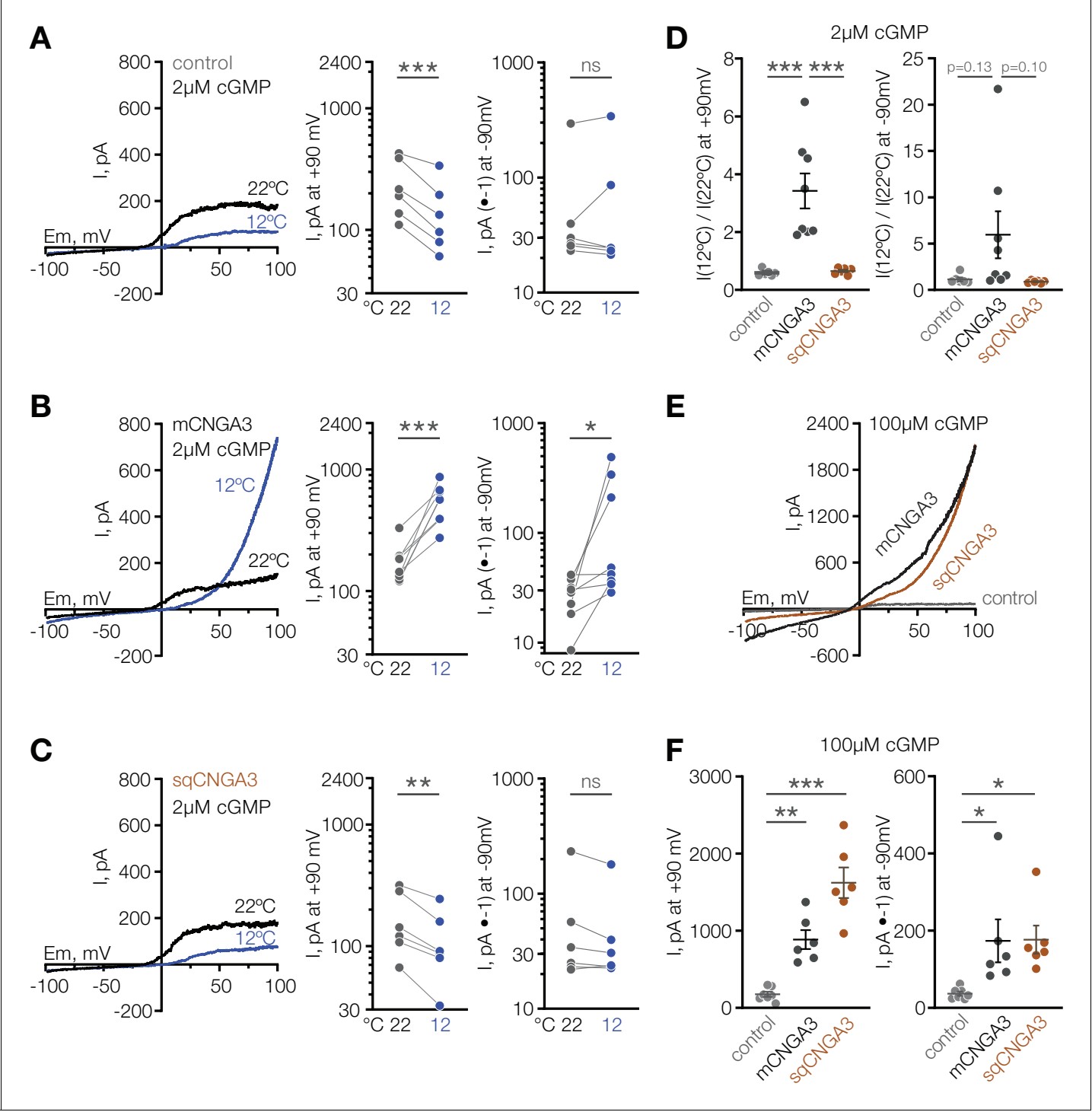

**Figure 7.** Cooling activates mouse but not squirrel CNGA3 in HEK293T cells. (A–C) Exemplar whole-cell currents (*left*) and quantification of current amplitude at 90 mV (*middle*) and −90 mV (*right*) recorded in HEK293T cells expressing empty vector (control) or CNGA3 in response to a voltage ramp in the presence of 2 μM intracellular cGMP, at indicated temperatures. Currents were elicited by voltage ramps from −100 mV to 100 mV from a holding potential of −60 mV. (D) Quantification of the magnitude of cold activation of CNGA3 current in HEK293T cells in the presence of 2 μM intracellular cGMP at 90 mV (*left*) and −90 mV (*right*). (E) Exemplar whole-cell current traces evoked in HEK293T cells expressing mCNGA3, sqCNGA3 or empty vector (control) by voltage ramps from −100 mV to 100 mV from a holding potential of −60 mV in the presence of saturating 100 μM intracellular cGMP at 22°C. (F) Quantification of current amplitude at 90 mV (*left*) and −90 mV (*right*) in the presence of saturating 100 μM intracellular cGMP at 22°C. Data are mean ± SEM. **p<0.01; ***p<0.001, paired t-test (A–C) or Dunnett's test (D, F). Each dot represents individual cell.

*Figure 7 continued on next page*

*Figure 7 continued*

The online version of this article includes the following source data and figure supplement(s) for figure 7:

**Source data 1.** Reversible activation of mouse CNGA3 in HEK293T cells.
**Figure supplement 1.** Reversible activation of mouse CNGA3 in HEK293T cells.

this, we investigated the effect of temperature on CNGA3 activity at different intracellular cGMP concentrations using inside-out patches of HEK293T cells (*Figure 9A*). Paired recordings from the same patches at different temperatures revealed a significant decrease in half-maximal cGMP concentration (EC$_{50}$) upon cooling from 21.8 ± 5.48 µM at 22°C to 3.5 ± 0.63 µM at 12°C (*Figure 9C and E*). In contrast to mouse CNGA3, and in agreement with our whole-cell data in oocytes and HEK293T cells, we did not detected a significant change in cGMP EC$_{50}$ for squirrel CNGA3 (26.4 ± 1.12 µM at 22°C, 20.4 ± 2.14 µM at 12°C, *Figure 9B,D and E*). Interestingly, cooling from 22°C to 12°C at saturating (≥100 µM) cGMP concentrations inhibited maximal activity of mouse CNGA3 by 25%, whereas this effect was significantly higher for the squirrel channel (60%, *Figure 9F*). Together, these findings reveal that the potentiating effect of cold at low cGMP concentrations is specific to mouse CNGA3, and is caused by a decrease in EC$_{50}$ for the cyclic nucleotide.

## Discussion

While progress has been made in identifying the markers and functional importance of warm-sensitive POA neurons (*Angilletta et al., 2019*; *Madden and Morrison, 2019*; *Siemens and Kamm, 2018*; *Tan and Knight, 2018*), the molecular basis and physiological role of cold sensitivity in the POA has remained obscure. Here, we report the identification of CNGA3 as a cold-potentiated ion channel in a subset of cold-sensing mouse POA neurons.

We established that cold decreases the effective concentration of cGMP needed for mouse CNGA3 activation, suggesting that it acts by enhancing the affinity between the channel and the nucleotide and/or facilitating channel opening in response to nucleotide binding (*James and Zagotta, 2018*). The observation that cold activates mouse CNGA3 in various cell types strongly suggests that this mechanism is intrinsic to the channel and does not involve cell type-specific

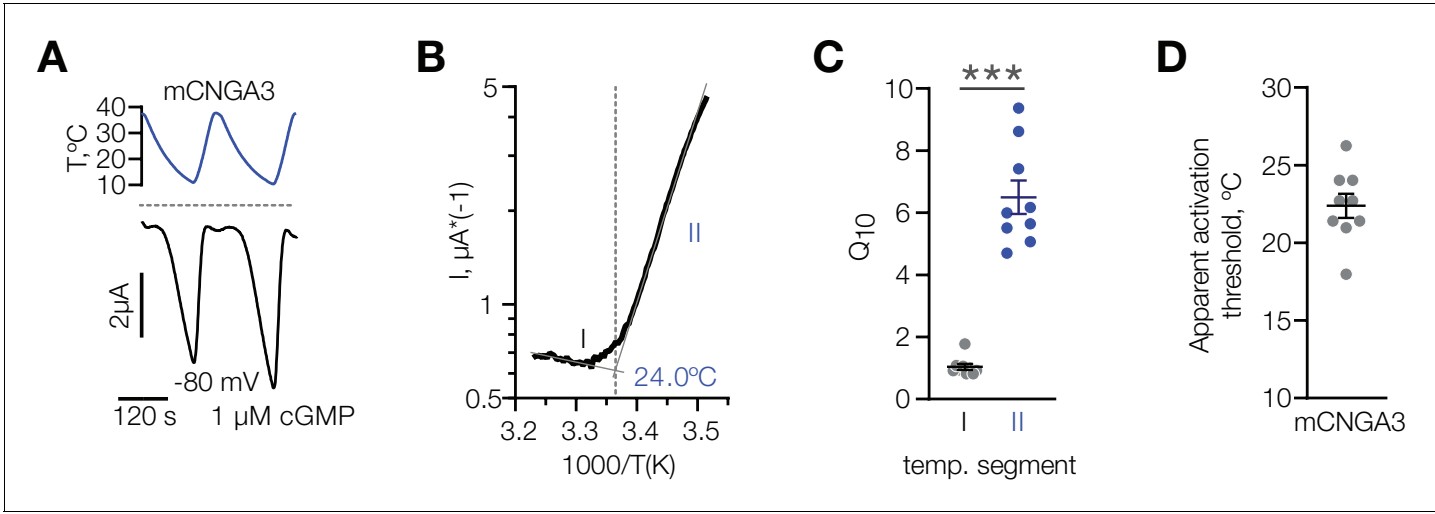

**Figure 8.** Characterization of cold sensitivity of mouse CNGA3. (**A**) Exemplar whole-cell currents obtained from *Xenopus* oocytes expressing mouse CNGA3 at −80 mV and 1 µM intracellular cGMP in response to temperature ramps from 37°C to 10°C. (**B**) Exemplar Arrhenius plot obtained from a recording as in (**A**) showing a bi-phasic temperature dependence of mCNGA3 current and apparent temperature activation threshold for this cell (24°C). (**C**) Quantification of the effect of cold on mCNGA3 current over 10°C (activation coefficient, [**Q$_{10}$**]). (**D**) Quantification of the apparent threshold of thermal activation of mCNGA3 from Arrhenius plots. Data are mean ± SEM; ***p<0.001, t-test.
The online version of this article includes the following source data for figure 8:

**Source data 1.** Characterization of cold sensitivity of mouse CNGA3.

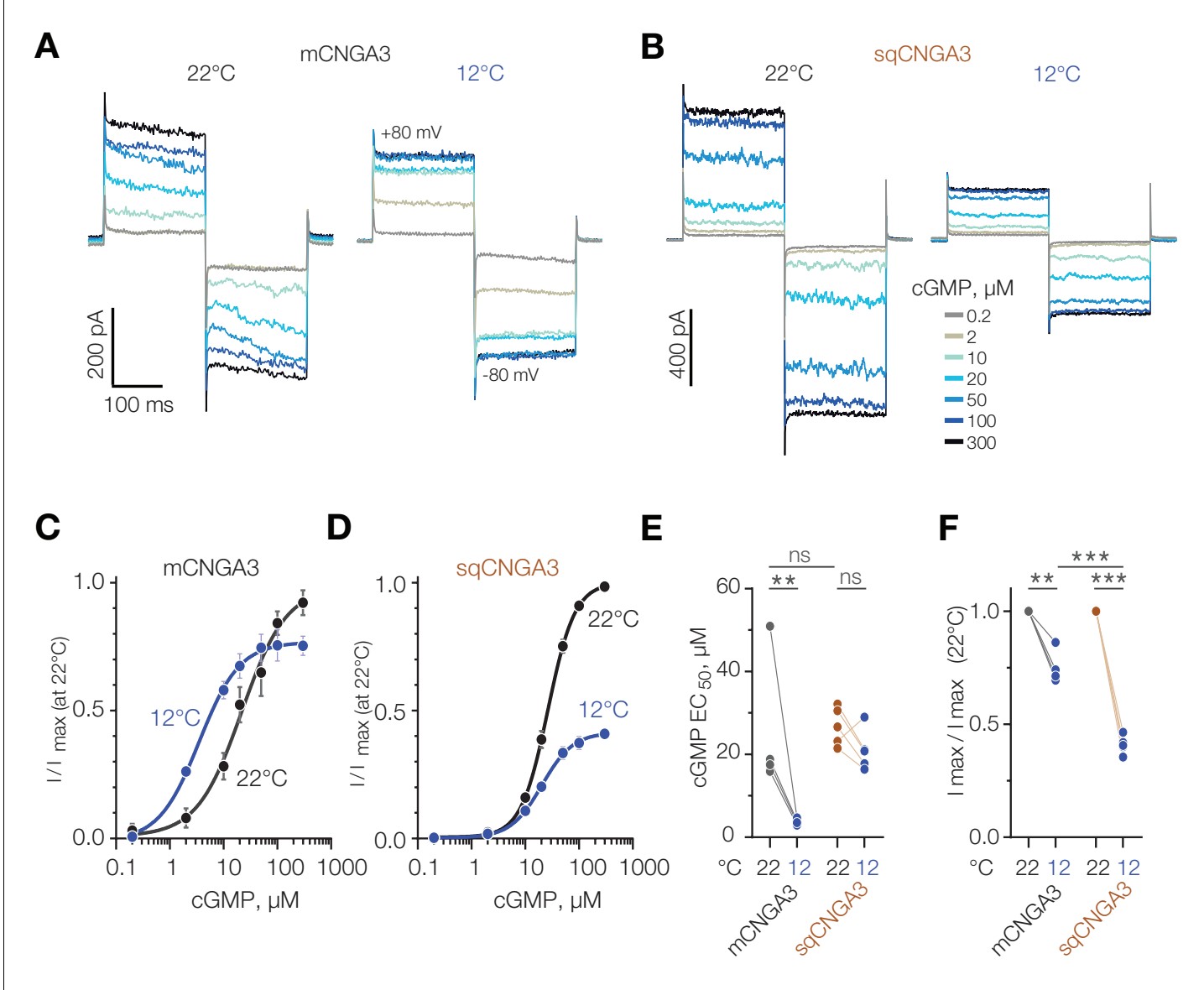

**Figure 9.** Cold decreases half-maximal effective concentration of cGMP for mouse CNGA3. (**A, B**) Exemplar current traces obtained in response to square voltage steps from inside-out patches of HEK293T cells expressing CNGA3. Each patch was tested at 22°C and 12°C at indicated 'intracellular' cGMP concentrations. (**C, D**) Concentration-dependence of CNGA3 on cGMP measured at 22°C and 12°C at −80 mV and normalized to maximal response at 22°C. Maximal responses were obtained by fitting the experimental data to a modified Hill equation (solid line). Data shown as mean ± SEM from 4 and 5 patches for mouse and squirrel CNGA3 respectively. (**E**) Quantification of half-maximal effective cGMP concentration ($EC_{50}$) for mouse and squirrel CNGA3 at different temperatures obtained from Hill fits in (**C**). Lines connect data points from individual patches measured at 22°C and 12°C. Statistical analysis: non-paired t-test (between 22°C data for different species), paired t-test (between 22°C and 12°C data pairs). (**F**) Quantification of maximal activity for mouse and squirrel CNGA3 at different temperatures obtained from Hill fits in (**D**). Lines connect data points from individual patches measured at 22°C and 12°C. Statistical analysis: non-paired t-test (between 12°C data for different species), paired t-test (between 22°C and 12°C data pairs). Error bars: mean ± SEM. 'ns', not significant, p>0.05; **p<0.01; ***p<0.001.

The online version of this article includes the following source data for figure 9:

**Source data 1.** Cold decreases half-maximal effective concentration of cGMP for mouse CNGA3.

components. Furthermore, our finding that cold acts on mouse CNGA3 even in inside-out patches, and fails to increase activity of mouse CNGA2 and the squirrel orthologue of CNGA3 argues against the involvement of indirect pathways such as cold-induced augmentation of cGMP production (*Chao et al., 2015*; *Mamasuew et al., 2010*; *Stebe et al., 2014*). A study in *C. elegans* showed that the cyclic nucleotide gated ion channel Tax-4 is expressed in thermosensory neurons, and that *tax-4*

mutants exhibit aberrant thermotaxic behavior, suggesting that thermosensitivity of mouse CNGA3 may have a deep evolutionary origin (*Komatsu et al., 1996*).

The physiological role of hypothalamic cold sensitivity is unclear. The strong correspondence between the proportion of cold-sensitive POA neurons and cold sensitivity of CNGA3 in both mice and squirrels suggests that the channel could be important for the detection of cold temperatures in the POA. A tantalizing possibility is that in mice the CNGA3-expressing cold-sensitive neurons could be a part of the mechanism that orchestrates thermogenesis in response to body cooling. In squirrels such mechanism is expected to be inhibited to allow for the profound drop of core body temperature during transition into torpor. Our observation that a small fraction of squirrel POA neurons exhibits cold sensitivity suggests the presence of a CNGA3-independent mechanism, which could be important for active monitoring of body temperature during hibernation (*Andrews, 2019*). Even though our transcriptome analysis did not reveal an enrichment for other known cold-sensors in mouse cold-sensitive compared to cold-insensitive neurons, we also do not exclude the existence of CNGA3-independent mechanisms of cold sensitivity in mouse POA.

Our data suggest that the CNGA3 sensor likely exists in POA neurons not only as a homomer, but also as a heteromer with CNGB1 (*Dai et al., 2013*; *Peng et al., 2004*; *Zhong et al., 2003*). Although the apparent activation threshold for homomeric CNGA3 is lower than the physiological brain temperature, it may depend on the cellular environment and recording conditions, similar to other thermo-sensitive channels (*Hoffstaetter et al., 2018*; *Song et al., 2016*; *Tan and McNaughton, 2016*; *Tominaga et al., 1998*). The temperature sensitivity of hypothalamic neurons is plastic and is modulated by metabolic state, circadian cycle, oxygenation, inflammatory status and peripheral nutrient signals (*Bartfai and Conti, 2012*; *Glotzbach and Heller, 1984*; *Pierau et al., 1998*; *Tattersall and Milsom, 2009*). In addition, the impact of CNGA3 on neuronal excitability will depend on the level of expression, neuronal resistance, and other cellular properties.

CNGA3 is best known for its role as a major cGMP-activated depolarizing ion channel in the cone photoreceptors (*Bönigk et al., 1993*; *Cukkemane et al., 2011*; *Dai et al., 2013*; *Kaupp and Seifert, 2002*; *Peng et al., 2004*; *Yu et al., 1996*). The channel is also implicated in the cold modulation of olfactory neurons in the Grueneberg ganglion, and is enriched in a subpopulation of peripheral cold receptors from dorsal root ganglia (*Luiz et al., 2019*; *Mamasuew et al., 2010*). Additionally, CNGA3 is expressed in the amygdala and hippocampus. Consistent with the expression pattern, a global deletion of *Cnga3* causes color blindness, affects odor recognition and produces deficits in hippocampal plasticity and amygdala-dependent fear memory (*Biel et al., 1999*; *Leinders-Zufall et al., 2007*; *Michalakis et al., 2011*). Given the complex phenotype of whole-body *Cnga3* knock-out animals, further understanding of the role of CNGA3 in thermoregulation would require a conditional ablation of the channel in the POA of adult mice and/or a targeted disruption of cold-sensing capability of CNGA3 while preserving its function as a cGMP sensor.

## Materials and methods

### Key resources table

| Reagent type (species) or resource | Designation | Source or reference | Identifiers | Additional information |
|---|---|---|---|---|
| Gene (*Mus musculus*) | *Cnga3* | This paper | GenBank MN381859 | Materials and methods |
| Gene (*Ictidomys tridecemlineatus*) | *Cnga3* | This paper | GenBank MN381860 | Materials and methods |
| Strain, strain background (*Mus musculus*) | C57Bl/6J | The Jackson Laboratory, Bar Harbor, ME | RRID: IMSR_JAX:000664 | Materials and methods |
| Strain, strain background (*Ictidomys tridecemlineatus*) | Thirteen-lined Ground Squirrel | Gracheva laboratory's colony | N/A | Materials and methods |

*Continued on next page*

*Continued*

| Reagent type (species) or resource | Designation | Source or reference | Identifiers | Additional information |
|---|---|---|---|---|
| Cell line (*H. sapiens*) | HEK293T$^{\Delta PIEZO1}$ | Dr. Ardem Patapoutian (Scripps Research Institute) (*Lukacs et al., 2015*) | N/A | Materials and methods |
| Sequence-based reagent | mouse CNGA3 cloning primer forward | his paper | N/A | 5'- GAGATGGCAAAGGTGAACAC −3' |
| Sequence-based reagent | mouse CNGA3 cloning primer reverse | this paper | N/A | 5'- GAGGCAGAGCCACCTGCATT −3' |
| Sequence-based reagent | ground squirrel CNGA3 cloning primer forward | this paper | N/A | 5'- GGACTGAATGCAACAAGCAAG −3' |
| Sequence-based reagent | ground squirrel CNGA3 cloning primer reverse | this paper | N/A | 5'- CCACGGAGCAGCTCATTTTC −3' |
| Commercial assay or kit | Quick-RNA Microprep Kit | Zymo, Irvine, Ca | Cat#: R1050 | Materials and methods |
| Commercial assay or kit | SMARTer Stranded Total RNA-Seq Kit - Pico Input Mammalian | Clontech/Takara Bio, Mountain View, CA | Cat#: 635005 | Materials and methods |
| Commercial assay or kit | RNAscope Multiplex Fluorescent Reagent Kit v2 Assay | Advanced Cell Diagnostics, Hayward, CA | Cat#: 323100 | Materials and methods |
| Commercial assay or kit | RNAScope mm Cnga3 probe | Advanced Cell Diagnostics, Hayward, CA | Cat#: 406131 | Materials and methods |
| Chemical compound, drug | cGMP-Na salt | Sigma | Cat#: G6129 | Materials and methods |
| Chemical compound, drug | L-cis-Diltiazem hydrochloride | Santa Cruz Biotechnology, Dallas, TX | Cat#: sc-221802 | Materials and methods |
| Software, algorithm | MetaFluor v7.8.2.0 | Molecular Devices, San Jose, CA | RRID:SCR_014294 | Materials and methods |
| Software, algorithm | ImageJ v1.51p | http://imagej.nih.gov/ij/; (*Schneider et al., 2012*) | RRID:SCR_003070 | Materials and methods |
| Software, algorithm | nd Stack Builder (ImageJ plugin) | https://imagej.nih.gov/ij/plugins/track/builder.html | N/A | Materials and methods |
| Software, algorithm | Trimmomatic v0.36 | (*Bolger et al., 2014*) | RRID:SCR_011848 | Materials and methods |
| Software, algorithm | STAR v2.5.4b | (*Dobin et al., 2013*) | RRID:SCR_015899 | Materials and methods |
| Software, algorithm | featureCounts v1.6.2 | (*Liao et al., 2014*) | RRID:SCR_012919 | Materials and methods |
| Software, algorithm | R v3.5.1 | https://www.r-project.org | RRID:SCR_001905 | Materials and methods |
| Software, algorithm | edgeR (package for R) v3.22.3 | (*Robinson et al., 2010*) | RRID:SCR_012802 | Materials and methods |
| Software, algorithm | pCLAMP v10 | Molecular Devices (https://www.moleculardevices.com/) | RRID:SCR_011323 | Materials and methods |
| Software, algorithm | GraphPad Prism v8.2.0 | GraphPad Software, San Diego, CA | RRID:SCR_002798 | Materials and methods |

## Contact for reagent and resource sharing

Further information and requests for resources and reagents should be directed to Elena Gracheva (elena.gracheva@yale.edu).

## Animals

All animal procedures were performed in compliance with the Office of Animal Research Support of Yale University (protocols 2018–11497 and 2018–11526). Thirteen-lined ground squirrels (*Ictidomys tridecemlineatus*), wild-type mice (*Mus musculus*), and frogs (*Xenopus laevis*) were used for this study. Wild-type C57Bl/6J mice were purchased from Jackson Laboratory (Bar Harbor, ME). All animals were housed on a 12 hr light/dark cycle (lights on at 0700) under standard laboratory conditions with ad libitum access to food and water. Both male and female mice 6–16 weeks of age weighing 17–34 g and male thirteen-lined ground squirrels 6 months-3 years of age weighing approximately 150–300 g were used for experiments. All ground squirrels were in their active (non-hibernating) state verified by daily body temperature measurements and maintained on a diet of dog food (Iams) supplemented with sunflower seeds, superworms, and fresh vegetables. Frogs were housed using standard conditions.

## Primary neuron dissociation from the POA of hypothalamus

Primary neurons were isolated from the POA of hypothalamus following a published protocol (*Vazirani et al., 2013*) with modifications. Mice or thirteen-lined ground squirrels were euthanized by isoflurane inhalation overdose followed by cardiac perfusion with the Brain Perfusion Solution (containing in mM: 196 sucrose, 2.5 KCl, 28 NaHCO3, 1.25 NaH2PO4, 7 Glucose, 1 Sodium Ascorbate, 0.5 CaCl2, 7 MgCl2, 3 Sodium Pyruvate, oxygenated with 95% O2/5% CO2, osmolarity adjusted to 300 mOsm with sucrose, pH adjusted to 7.4). Brain was dissected and brain slices were cut on a vibratome (VT1200, Leica Biosystems Inc, Buffalo Grove, IL). A brain slice containing the POA was identified by the presence of the anterior commissure crossover, and 2 successive 300 µm slices containing the POA were collected. A square region approximately 2 × 3 mm between the optic chiasm and anterior commissure crossover containing the POA was microdissected from the brain slices. Tissue was digested in Hibernate A medium (custom formulation with 2.5 mM glucose and osmolarity adjusted to 280 mOsm, BrainBits, Springfield, IL) supplemented with 1 mM lactic acid (Cat. #L1750, Sigma, St. Louis, MO), 0.5 mM GlutaMAX (Cat. #35050061, ThermoFisher, Waltham, MA) and 2% B27 minus insulin (Cat. #A1895601, ThermoFisher) containing 20 U/ml papain (LS003124, Worthington Biochemical Corporation, Lakewood, NJ) in a shaking water bath at 34°C for 30 min and dissociated by mechanical trituration through the tips of glass Pasteur pipettes with decreasing diameter. Cell suspension was centrifuged over 8% bovine serum albumin (A9418-5G, Sigma, St. Louis, MO) layer, resuspended in Neurobasal-A medium (A2477501 [no glucose, no sodium pyruvate], custom formulation with 2.5 mM glucose and osmolarity adjusted to 280 mOsm), supplemented with 1 mM lactic acid, 0.5 mM GlutaMAX and 2% B27 minus insulin, plated on poly-D-lysine/laminin-coated glass coverslips (Cat. #354087, Corning Inc, Corning, NY) and cultured in an incubator at 34°C in 5% CO2. Plated POA neurons were used for experiments within 24 hr.

## Live-cell ratiometric calcium imaging

POA neurons acutely cultured on glass coverslips were loaded with 10 µM Fura 2-AM (Cat. #F1201, ThermoFisher) and 0.02% Pluronic F-127 (Cat. #P3000MP, ThermoFisher) in the Recording Solution (*Vazirani et al., 2013*) (in mM: 121 NaCl, 4.7 KCl, 2.5 D-glucose, 5 NaHCO3, 2 CaCl2, 0.1 MgCl2, 1.2 MgSO4, 0.97 KH2PO4, 0.23 K2HPO4, 25 HEPES, Osmolarity 280 mOsm, pH 7.4) for 30 min at 34°C and washed three times with the Recording Solution. Live-cell ratiometric calcium imaging was performed using the Axio-Observer Z1 inverted microscope (Carl Zeiss Inc, Thornwood, NY) equipped with the Orca-Flash4.0 camera (Hamamatsu, Bridgewater, NJ) using Meta-Fluor software v7.8.2.0 (Molecular Devices, San Jose, CA). Fluorescent images at 340 and 380 nm excitation were obtained with 10x objective every 1 s. Exemplar ratiometric images (*Figure 1A*) are presented using the Meta-Fluor Intensity Modulated Display color coding. Cells were continuously perfused with the Recording Solution at a flow rate of ~5 ml/min. After obtaining baseline values at room temperature for 50 s, a cooling or a warming ramp was applied by perfusing the cells with the Recording Solution passed through the in-line Peltier heater-cooler (SC-20, Warner Instruments, Hamden, CT)

connected to the Dual Channel Bipolar Temperature Controller (CL-200A, Warner Instruments). Temperature in the bath was recorded using a bead thermistor (TA-29, Warner Instruments). To produce a cooling ramp, cooled solution was delivered for 30 s, achieving the temperature of ~10˚C in the bath. To produce a warming ramp, warmed solution was delivered to the bath until the temperature in the bath measured by thermistor reached 48˚C. At the end of the recording, cells were perfused with high-potassium solution ('High-K$^+$' in figures, in mM: 10 NaCl, 115.7 KCl, 2.5 D-glucose, 5 NaHCO3, 2 CaCl2, 0.1 MgCl2, 1.2 MgSO4, 0.97 KH2PO4, 0.23 K2HPO4, 25 HEPES, Osmolarity 280 mOsm, pH 7.4) to identify functionally intact neurons. In experiments with pharmacological inhibition of CNGA3, l-cis-diltiazem hydrochloride (sc-221802, Santa Cruz Biotechnology, Dallas, TX) was added to the recording solution at a final concentration of 50 µM from 10 mM stock prepared in water.

## Data analysis of calcium imaging data

Fluorescent images were reprocessed offline. Regions of interest over imaged cells were created using the automated segmentation feature in the MetaFluor software and manually revised as needed. An exclusive threshold was applied to the 340nm-excitation image to exclude regions not containing cells from calculations. The average 340/380 nm excitation ratio values over each region of interest were calculated over the time course of the recording and exported to a Microsoft Excel worksheet. A custom Excel macro was applied to process the data. Only cells passing the following empirically determined quality control criteria were included in further analysis: the average baseline (defined as the first 45 s of the recording) F ratio $\leq$ 0.7 fluorescence ratio units, baseline standard deviation $\leq$ 0.05 units, the difference between the pre-High-K$^+$ epoch (20 s prior to the High-K$^+$ solution application) and baseline $\leq$ 0.5 units (ensuring the recovery of the signal after cooling or warming), the High-K$^+$ response amplitude $\geq$0.5 units (defined as the difference between the maximal High-K$^+$ and the average pre-High-K$^+$fluorescence ratio value and denoting a functional response to depolarizing high potassium solution characteristic of healthy neurons). The amplitude of the response to cooling or warming for each cell passing quality control criteria was determined as the difference between the maximal fluorescent ratio value during the cooling or warming ramp (plus 10 s allowing for a delayed response) and the average baseline fluorescence ratio. A neuron was defined as cold-sensitive or warm-sensitive if the amplitude of the response to cooling or warming respectively was greater than or equal to 0.575 fluorescence ratio units. The percentage of cold- or warm-sensitive neurons was determined for each coverslip (containing ~50–300 neurons) and averaged over all coverslips within each experimental condition. Each experiment included 3–6 coverslips from 6 to 13 animals/independent neuron isolation procedures.

## Cell collection

Following the identification of cold-sensitive POA neurons by live-cell ratiometric calcium imaging, single cold-sensitive and -insensitive neurons were separately collected and pooled for subsequent transcriptomic analysis. These experiments were conducted in RNAse-free conditions. A custom Microsoft Excel macro and a modified ImageJ plugin (https://imagej.nih.gov/ij/plugins/track/builder.html) were used to create an annotated bright-field image with neurons marked based on cold sensitivity values. Cold-sensitive neurons, defined as those in the top 7% by cold response amplitude, as well as cold-insensitive neurons, in the bottom 14%, were separately targeted for collection. An aspiration pipette was pulled from capillary glass tubing (G150F-3, Warner Instruments, Hamden, CT) using a micropipette puller (P-1000, Sutter, Novato, CA) with a tip diameter of ~20–40 µM, filled with 3 µl of the RNA Lysis Buffer (Quick-RNA Microprep Kit, Zymo, Irvine, Ca), loaded into micromanipulator, connected to a 1 ml syringe for suction application, and used to aspirate 1–10 cells from a coverslip. Collected neurons were then deposited into a 0.5 ml tube containing 10 µl of the RNA Lysis Buffer. Cell collection was repeated with each coverslip (5–6 total per one neuron isolation procedure) using a different aspiration pipette and collected cells of the same type (cold-sensitive and -insensitive) from each coverslip were pooled together in one tube. Samples were then stored at −80˚C until RNA isolation. The procedure was repeated with 2–6 independent neuronal isolations and cell collection sessions until ~ 100–200 cells were collected to obtain one biological replicate, with a total of 3 biological replicates. All samples of one type within each biological replicate were then pooled together and RNA was isolated using the Quick-RNA Microprep Kit (Zymo) according

to manufacturer's instructions. RNA concentration and integrity number (RIN) were assessed by Agilent 2100 Bioanalyzer (Agilent, Santa Clara, CA). RNA concentrations were in the range of 58–211 pg/µl and RIN values were in the range of 4.0–8.9.

## RNA sequencing

Library preparation and sequencing were carried out at the Yale Center for Genome Analysis. Sequencing libraries were prepared using the SMARTer Stranded Total RNA-Seq Kit - Pico Input Mammalian (Cat. #635005, Clontech/Takara Bio, Mountain View, CA) including rRNA depletion. Libraries were sequenced on Illumina HiSeq 2500 in the 75 bp paired-end mode according to manufacturer's protocols with four samples pooled per lane. A total of ~ 23–31 million sequencing read pairs per sample were obtained.

The sequencing data was processed on the Yale Center for Research Computing cluster. Raw sequencing reads were filtered and trimmed to retain high-quality reads using Trimmomatic v0.36 (*Bolger et al., 2014*) with default parameters. Filtered high-quality reads from all samples were aligned to mouse reference genome using the STAR aligner v2.5.4b with default parameters (*Dobin et al., 2013*). The reference genome and the gene annotation were obtained from the Gencode project (*Frankish et al., 2019*) (Release M16 [GRCm38.p6]; all files accessed on 2/23/2018).

### Reference genome

ftp://ftp.ebi.ac.uk/pub/databases/gencode/Gencode_mouse/release_M16/GRCm38.primary_assembly.genome.fa.gz;

### Gene annotation

ftp://ftp.ebi.ac.uk/pub/databases/gencode/Gencode_mouse/release_M16/gencode.vM16.annotation.gff3.gz;

The gene annotation was filtered to include only protein-coding genes. Aligned reads were counted by featureCounts program within the Subread package v1.6.2 with default parameters (*Liao et al., 2014*). Read counting was performed at the gene level, i.e. the final read count for each gene included all reads mapped to all exons of this gene. Differential expression of genes based on read counts and fold-change between cold-sensitive and cold-insensitive neurons was determined by EdgeR v3.22.3 (*Robinson et al., 2010*) using the following parameters: features with zero counts in at least one sample were excluded prior to statistical analysis; statistical analysis was performed using the GLM approach and the quasi-likelihood F-test; features with less than one normalized read/million in at least one sample were excluded post statistical analysis; features were filtered to include only those with the adjusted p-value≤0.05 and sorted by the largest fold-change. Normalized read counts were obtained by normalizing raw read counts to effective library sizes of each sample and expressed as reads/million of total reads in a library. Effective library sizes were calculated by normalizing raw library sizes by RNA composition using a trimmed mean of M-values (TMM) method, as implemented in calcNormFactors function of the EdgeR package. To obtain normalized read counts for *Cnga2* and *Cngb3* (which had zero read counts in some samples), the EdgeR analysis described above was repeated without filtering out features with zero counts.

## RNA in situ hybridization

*Cnga3* mRNA expression was detected in mouse brain by RNA in situ hybridization using the RNAscope Multiplex Fluorescent Reagent Kit v2 Assay (Cat #323100, Advanced Cell Diagnostics, Hayward, CA) according to manufacturer's instructions. Mice were euthanized and transcardially perfused with ice-cold 4% paraformaldehyde in PBS, brain was dissected, fixed for 24 hr in 4% paraformaldehyde at 4°C on a rocker platform, dehydrated in 10, 20, and 30% sucrose solution in PBS (pH = 7.4) successively at 4°C on a rocker platform until sunk, frozen in Tissue-Tek O.C.T. compound (Cat. #62550–01, Electron Microscopy Sciences, Hatfield, PA) and stored at −80°C. Brain harvest was performed in strict RNA-se free conditions. The brain tissue block was cut on a cryostat (CM3500S, Leica Biosystems Inc, Buffalo Grove, IL) into 14 µm sections. A brain section containing the anterior commissure crossover was used as a landmark for the POA (corresponding to the anteroposterior brain coordinate of bregma +0.14 mm, according to the electronic brain atlas (http://labs.gaidi.ca/

mouse-brain-atlas/) based on *Paxinos and Franklin (2001)*, and the sections covering the next distal 600 µm were collected for processing. Representative sections shown in *Figure 2* correspond to the approximate AP coordinate of bregma −0.2 mm. Brain sections were mounted on glass slides (SuperFrost Plus, Cat. #12-550-15, Fisher Scientific, Pittsburgh, PA), air dried for 1 hr at room temperature (RT), washed with PBS, baked at 60°C for 30 min, post-fixed in 4% paraformaldehyde for 90 min at RT, dehydrated by successive incubation in 50, 70, and 100% (twice) ethanol for 5 min at RT, and air dried for 5 min. Sections were then processed and hybridized to RNA probes according to the RNAscope kit instructions. For *Cnga3* detection, sections were incubated with the RNAScope *mCnga3* (in C1 channel) or 3-plex Negative Control probe against the bacterial dapB gene (Cat. #406131 and #320871 respectively, Advanced Cell Diagnostics), followed by incubation in TSA Plus Cyanine 5 detection reagent (Cat. #NEL745E001KT, PerkinElmer, Waltham, MA). Brain sections were counterstained with DAPI. Processed brain sections were imaged on a Zeiss 700 confocal microscope (Carl Zeiss Inc, Thornwood, NY) using 405 nm (DAPI) and 633 nm (Cy5) laser lines and 63X oil objective. Z-stacks containing 6 images at 1 µm Z-step were collected and maximal intensity projection images were constructed. Images presented in *Figure 1G* are representative of 8 fields-of-view from 4 sections from two independent procedures.

## Plasmids
Mouse *Cnga2* (NCBI accession number: NM_007724.3) and *Cngb1* (NCBI accession number: NM_001195413.1) in pcDNA3.1+/C-(K)DYK vector were obtained from a commercial source (CloneID: OMu19184 and OMu17438 respectively, GenScript, Piscataway, NJ) and subcloned into the pMO vector.

## *Cnga3* gene cloning
Mouse and ground squirrel orthologues of the *Cnga3* gene were cloned from the POA of respective species using standard techniques and the following primers: Mouse forward 5'- GAGA TGGCAAAGGTGAACAC −3'; reverse 5'- GAGGCAGAGCCACCTGCATT −3'; Ground squirrel: forward 5'- GGACTGAATGCAACAAGCAAG −3'; reverse 5'- CCACGGAGCAGCTCATTTTC −3'. ORFs were then subcloned into the pMO vector for expression in HEK cells and *Xenopus* oocytes. All constructs were verified by full-length sequencing. The nucleotide and protein sequences of the cloned transcripts were deposited to GenBank under the accession numbers: MN381859 (mouse *Cnga3*), MN381860 (ground squirrel *Cnga3*). The protein sequences are shown below.

## Mouse CNGA3
Our cloned variant (630 amino acids) is identical to the NCBI reference sequence XP_017169237 (630 amino acids).

```
MAKVNTQCSQPSPTQLSIKNADRDLDHVENGLGRVSRLIISIRAWASRHL
HDEDQTPDSFLDRFHGSELKEVSTRESNAQPNPGEQKPPDGGEGKEEPIV
VDPSSNIYYRWLTAIALPVFYNWCLLVCRACFDELQSEHLTLWLVLDYSA
DVLYVLDMLVRARTGFLEQGLMVRDTKRLWKHYTKTLHFKLDILSLIPTD
LAYLKLGVNYPELRFNRLLKFSRLFEFFDRTETRTNYPNVFRIGNLVLYT
LIIIHWNACIYFAISKFIGFGTDSWVYPNTSKPEYARLSRKYIYSLYWST
LTLTTIGETPPPVKDEEYLFVVIDFLVGILIFATIVGNVGSMISNMNAPR
VEFQAKIDSVKQYMQFRKVTKDLETRVIRWFDYLWANRKTVDEKEVLKNL
PDKLKAEIAINVHLDTLKKVRIFQDCEAGLLVELVLKLRPTVFSPGDYIC
KKGDIGREMYIIKEGKLAVVADDGVTQFVVLSDGSYFGEISILNIKGSKS
GNRRTANIRSIGYSDLFCLSKDDLMEALTEYPDAKRALEEKGRQILMKDN
LIDEDLVAARVDTRDVEEKVEYLESSLDILQTRFARLLAEYSASQMKLKQ
```

## Thirteen-lined ground squirrel CNGA3
Our cloned variant (672 amino acids) differs from the NCBI reference sequence (XP_005339657, 711 amino acids) by lacking a region between positions 132–170, and having one substitution, glutamic acid to arginine in position 171.

```
MAKVSTQYSRPSLTHLPTKTVDRDLDRAENGLSRGHLPCEETPTALQQGI
AMETREPAGPPQSSFTGQGPARLARLIISLRTWTARRSRCEDQRSDSPPD
```

```
RFRGAELKEVSSQESNAQSHAGSQEPPDRGRRKKKESFVMDPSSNLYYRW
LTTIAVPVFYNWCLLVCRACFDELQSEHLMLWLVLDYSSDVIYGLDMLVR
TRTGFLEQGLMVQDTSRLWKHYTKSMQFKLDVLSLVPTDLAYIKWGMNYP
ELRFNRLLRLSRLFEFFDRTETRTSYPNVFRIGNLVLYILVIIHWNACIY
FAISKFIGFGTDSWVYPNISKPEHARLSRKYIYSLYWSTLTLTTIGETPP
PVKDGEYLFVVIDFLVGVLIFATIVGNVGSMISNMNASRAEFQAKIDSIK
QYMQFRKVTKDLETRVIRWFDYLWANGKTVDEKEVLKSLPDKLKAEIAIN
VHLDTLRKVRIFQDCEAGLLVELVLKLRPTVFSPGDYICKKGDIGKEMYI
IKEGKLAVVADDGITQFVVLSDGSYFGEISILNIKGSKSGNRRTANIRSI
GYSDLFCLSKDDLMEALTEYPEAKKALEEKGRQILMKDNLIDEDVAKAGA
DPKDIEEKVEHLESSLDMLQTRFARLLAEYNTNQMKVKQRLSQLESQVKG
SGSGPPSDAEAPEEAAKTEAKQ
```

## Two-electrode voltage clamp electrophysiology in oocytes

Two-electrode voltage clamp electrophysiology was performed on defolliculated stage V-VI *Xenopus laevis* oocytes microinjected with 0.5–6.0 ng cRNA encoding CNGA3, CNGA2 or CNGB1 synthesized by in vitro transcription from linearized plasmids using the mMessage mMachine kit (Cat. #AM1344, ThermoFisher). Oocytes were cultured in ND96 media (in mM): 96 NaCl, 2 KCl, 2 MgCl$_2$, 1.8 CaCl$_2$, 10 HEPES, pH 7.4 with NaOH) for 24–72 hr prior to recordings. Whole-cell recordings were performed using borosilicate glass microelectrodes (0.5–2.5 MΩ resistance) filled with 3M KCl, under continuous perfusion with ND96 without calcium to avoid contamination with endogenous calcium-activated chloride current (ND96 without Ca$^{2+}$, in mM: 96 NaCl, 2 KCl, 2 MgCl$_2$, 10 HEPES, pH 7.4 with NaOH), using the OC-725 amplifier (Warner Instruments) and pClamp 10.3 software suite (Molecular Devices). Currents were evoked by a 2 s voltage ramp from −120 mV to 60 mV from a holding potential of −80 mV, filtered at 1 kHz and sampled at 5 kHz using the Digidata 1440A digitizer (Molecular Devices). Alternatively, currents were recorded using a gap-free protocol at −80 mV, lowpass-filtered at 1 kHz and sampled at 2 kHz. For gap-free recordings, oocytes were injected with higher amount of cRNA and/or incubated longer than for voltage ramp recordings to achieve high current amplitude. To deliver cGMP into oocytes, cells were microinjected with 50 µl of 20 µM or 5 mM cGMP 90 s prior to recording to achieve, respectively, 1 µM or 250 µM final intracellular concentration, assuming the average oocyte volume at 1 µl. Unless indicated otherwise, recordings were started 1.5–2 min post-injection to allow cGMP to diffuse inside the oocyte. Temperature of the extracellular solution was controlled using the SC-20 in-line heater-cooler and CL-100 temperature controller (Warner Instruments) and monitored using a thermistor placed adjacent to the oocyte.

## Patch-clamp electrophysiology in HEK293T cells

Electrophysiological recordings were performed in HEK293T cells with genomic deletion of *PIEZO1*, a kind gift by Ardem Patapoutian (Scripps Research Institute), were tested negative for mycoplasma and authenticated by PCR and short-tandem-repeat analysis (*Lukacs et al., 2015*). Cells were cultured in DMEM+ media (DMEM with 10% FBS, 1% Penicillin/Streptomycin, and 2 mM glutamine) using standard procedures, transfected with 2–5 µg pMO vector-based plasmids encoding mouse or squirrel CNGA3 together with 0.1–0.25 µg pcDNA3.1-eGFP plasmids using the Lipofectamine 3000 reagent (Thermo) in Opti-MEM (Gibco), plated onto coverslips coated with Matrigel (BD Bioscience) and recorded within 24–48 hr after transfection. Whole-cell recordings were performed in extracellular solution (in mM: 140 NaCl, 5 KCl, 2.5 CaCl$_2$, 1 MgCl$_2$, 10 glucose, 10 HEPES, pH 7.2 with NaOH) using 1–3 MΩ resistance electrodes filled with intracellular solution (in mM: 150 KCl, 3 MgCl2, 5 EGTA, 10 HEPES, pH 7.2 with KOH, supplemented or not with 2 µM or 100 µM cGMP) using the Axopatch 200B amplifier and pClamp 10.3 software suite (Molecular Devices). Currents were evoked by a 1 s voltage ramp from −100 mV to 100 mV, from a holding potential of −60 mV, filtered at 2 kHz and sampled at 5 kHz using the Digidata 1440A digitizer (Molecular Devices). Inside-out patch recordings were performed using 0.8–1.5 MΩ resistance electrodes filled with Na-EDTA solution (in mM: 130 NaCl, 0.2 EGTA, 5 HEPES, pH 7.2 with NaOH). The 'intracellular' bath solution contained Na-EDTA supplemented with cGMP. Currents were elicited by two 400 ms voltage steps to +80 mV and −80 mV from a holding potential of 0 mV, filtered at 1 kHz and sampled at 5 kHz. Concentration-dependence curves for CNGA3 current were obtained by fitting experimental data to the

modified Hill equation: $I = I_{min}+(I_{max}-I_{min})/(1+(EC_{50}/[cGMP])^H)$, where I is the baseline-subtracted CNGA3 current measured at a specified cGMP concentration ([cGMP]), $I_{min}$ and $I_{max}$ are minimal and maximal projected current values, $EC_{50}$ is the half-maximal effective concentration of cGMP, and H is the Hill coefficient (slope). Temperature of the solution was controlled using the SC-20 in-line heater-cooler and CL-100 temperature controller (Warner Instruments) and monitored using a thermistor placed in the recording chamber.

## Statistical analysis

Statistical analyses were performed in GraphPad Prism v8.2.0 (GraphPad Software, San Diego, CA) and R v3.5.1. Data were tested for normality using the Kolmogorov-Smirnov or Shapiro-Wilk test. Welch's t-test and 1-way ANOVA followed by Dunnett's pairwise multiple comparison test were used for normally distributed data, and Mann-Whitney test was used for non-normally distributed data. Statistical analysis of the RNA sequencing data was performed by the built-in statistical tools of the EdgeR package as described in the respective section. Tests used for individual experiments and sample sizes are reported in the Results section and in figure legends. Data in the text and graphs were reported as mean ± SEM. The p-values associated with statistical tests were reported as 'ns' ($p>0.05$), '*' ($p<0.05$), '**' ($p<0.01$), and '***' ($p<0.001$). The differences were considered statistically significant at $p<0.05$.

## Acknowledgements

We thank Thomas McCabe, Lyle Murphy and Jon Matson for technical assistance, and members of the SNB and EOG laboratories for their contributions throughout the project. VVF was supported by the James Hudson Brown - Alexander B Coxe postdoctoral fellowship. This study was partly funded by NSF grant 1754286 and NIH grant 1R01NS091300-01A1 (to EOG) and by NSF grant 1923127 and NIH grant 1R01NS097547-01A1 (to SNB).

## Additional information

### Funding

| Funder | Grant reference number | Author |
|---|---|---|
| Yale University | James Hudson Brown - Alexander B Coxe Postdoctoral fellowship | Viktor V Feketa |
| National Science Foundation | 1754286 | Elena O Gracheva |
| National Institute of Neurological Disorders and Stroke | 1R01NS091300-01A1 | Elena O Gracheva |
| National Science Foundation | 1923127 | Sviatoslav N Bagriantsev |
| National Institute of Neurological Disorders and Stroke | 1R01NS097547-01A1 | Sviatoslav N Bagriantsev |

The funders had no role in study design, data collection and interpretation, or the decision to submit the work for publication.

### Author contributions

Viktor V Feketa, Conceptualization, Data curation, Software, Formal analysis, Validation, Investigation, Visualization, Methodology, Writing - original draft, Writing - review and editing; Yury A Nikolaev, Data curation, Formal analysis, Investigation, Visualization, Writing - review and editing; Dana K Merriman, Writing - review and editing, Supplied squirrels and provided advice on animal husbandry; Sviatoslav N Bagriantsev, Elena O Gracheva, Conceptualization, Resources, Data curation, Formal analysis, Supervision, Funding acquisition, Validation, Investigation, Visualization, Methodology, Writing - original draft, Project administration, Writing - review and editing

### Author ORCIDs

Viktor V Feketa (iD) https://orcid.org/0000-0003-4978-0581
Sviatoslav N Bagriantsev (iD) https://orcid.org/0000-0002-6661-3403
Elena O Gracheva (iD) https://orcid.org/0000-0002-0846-3427

### Ethics

Animal experimentation: All animal procedures were performed in compliance with the Office of Animal Research Support of Yale University (protocols 2018-11497 and 2018-11526). Thirteen-lined ground squirrels (Ictidomys tridecemlineatus), wild-type mice (*Mus musculus*), and frogs (*Xenopus laevis*) were used for this study. Wild-type C57Bl/6J mice were purchased from Jackson Laboratory (Bar Harbor, ME). All animals were housed on a 12-h light/dark cycle (lights on at 0700) under standard laboratory conditions with ad libitum access to food and water. Both male and female mice 6-16 weeks of age weighing 17-34 g and male thirteen-lined ground squirrels 6 months-3 years of age weighing approximately 150-300 g were used for experiments. All ground squirrels were in their active (non-hibernating) state verified by daily body temperature measurements and maintained on a diet of dog food (Iams) supplemented with sunflower seeds, superworms, and fresh vegetables. Frogs were housed using standard conditions.

### Decision letter and Author response

Decision letter https://doi.org/10.7554/eLife.55370.sa1
Author response https://doi.org/10.7554/eLife.55370.sa2

## Additional files

### Supplementary files

• Transparent reporting form

### Data availability

The RNA sequencing data was deposited to the Gene Expression Omnibus, accession number: GSE136396. The nucleotide and protein sequences of the cloned mouse and ground squirrel CNGA3 orthologues were deposited to GenBank under the accession numbers: MN381859 (mouse Cnga3), MN381860 (ground squirrel Cnga3).

The following datasets were generated:

| Author(s) | Year | Dataset title | Dataset URL | Database and Identifier |
|---|---|---|---|---|
| Feketa V, Bagriantsev S, Gracheva E | 2020 | RNAseq of cold-sensitive and cold-insensitive neurons from the preoptic area of hypothalamus of mice | https://www.ncbi.nlm.nih.gov/geo/query/acc.cgi?acc=GSE136396 | NCBI Gene Expression Omnibus, GSE136396 |
| Feketa VV, Nikolaev YA, Merriman DK, Bagriantsev SN, Gracheva EO | 2020 | Mus musculus strain C57BL/6J cyclic nucleotide-gated channel subunit alpha 3 (Cnga3) mRNA, complete cds | https://www.ncbi.nlm.nih.gov/nuccore/MN381859 | NCBI GenBank, MN381859 |
| Feketa VV, Nikolaev YA, Merriman DK, Bagriantsev SN, Gracheva EO | 2020 | Ictidomys tridecemlineatus cyclic nucleotide-gated channel subunit alpha 3 (Cnga3) mRNA, complete cds | https://www.ncbi.nlm.nih.gov/nuccore/MN381860 | NCBI GenBank, MN381860 |

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
