## [Decision Letter]

**Acceptance summary:**

This meticulously and artfully executed study on the molecular mechanisms of thermoregulation revealed an unexpected new player, the cyclic nucleotide-gated ion channel CNGA3, that acts as a cold-regulated ion channel in murine hypothalamus. The discovery adds to the molecular diversity of the temperature-sensitive elements and opens new areas of investigation. Since CNGA3 is expressed in the cone photoreceptors, olfactory neurons, as well as nonneuronal tissues, such as kidney and heart, it may contribute to the local thermoregulation of these organs, and represents a fertile area for future studies.

**Decision letter after peer review:**

Thank you for submitting your article "CNGA3 is a cold sensor in hypothalamic neurons" for consideration by *eLife*. Your article has been reviewed by three peer reviewers, and the evaluation has been overseen by a Reviewing Editor and Richard Aldrich as the Senior Editor. The following individuals involved in review of your submission have agreed to reveal their identity: Makoto Tominaga (Reviewer #2); Alexander Sobolevsky (Reviewer #3).

The reviewers have discussed the reviews with one another and the Reviewing Editor has drafted this decision to help you prepare a revised submission. Our goal is to provide the essential revision requirements as a single set of instructions so that you have a clear view of the revisions that are necessary for us to publish your work.

Summary:

All three reviewers were very impressed with the quality of the work and highly recommend it for publication. The reviewers find that the manuscript presents a very exciting and rigorous examination of a novel hypothalamic cold sensor. The authors show that mice have larger numbers of cold-sensitive neurons in the preoptic area (POA) than hibernating ground squirrels, and demonstrate upregulation of the cyclic nucleotide-gated ion channel CNGA3 in cold-sensitive neurons in mice. The study is well conducted and the manuscript is clear. Moreover, the results are significant since they show for the first time a new mechanism of cold detection in POA neurons.

Central conclusions:

1) Feketa et al. discovered that cold-sensitive neurons in POA have a highly expressed subtype of CNG channels, specifically CNGA3, which responds to cool temperature in mice but not in hibernating ground squirrels.

2) The authors determined that the temperature threshold for murine CNGA3 and the temperature coefficient are higher than that for an average protein. Hence, the authors suggested that murine CNGA3 may play a role as a cold temperature sensor.

3) This study highlights the strength of using a comparative approach to address important questions in neuroscience and physiology and has broad relevance for the neurobiology of thermosensation, hibernation biology, as well as for the study of protein evolution. Moreover, as reviewers point out, the discovery of orthologs with different properties can enable subsequent structure-function studies.

Essential revisions:

The reviewers provided a small number of corrections and suggestions that must be adequately addressed before the paper can be formally accepted. One of the suggested revisions is to provide a comparison of the dose-response curve to cGMP between mouse and ground squirrel at a negative potential.

1) CNGA3 is not per se a real cold sensor, rather the channel becomes sensitive to low cGMP levels below a certain temperature threshold. One might argue about the appellation of a cold sensor, as cold acts more like a subthreshold amplifier of the channel function.

2) In Figure 1, authors observed a subpopulation of cold-sensitive neurons that represented 10% and 6% of mouse and ground squirrel total POA neurons population, respectively. The difference, although statistically different, is very minor and suggest that another cold sensor might be expressed in ground squirrel POA neurons. Brain temperature of hibernating ground squirrel can be as low as 5ºC. How do the authors reconcile this with their observation that ground squirrel POA neurons can still respond to cold?

3) In Figure 4F authors show that there is no difference for cGMP sensitivity between mouse and ground squirrel CNGA3 in HEK293 cells at +90 mV. However, there seems to be a difference at negative potentials, and in an opposite fashion of what is observed in oocytes. A comparison of the dose-response curve to cGMP between mouse and ground squirrel at negative potential would be more convincing.

---

## [Author Response]

Essential revisions:The reviewers provided a small number of corrections and suggestions that must be adequately addressed before the paper can be formally accepted. One of the suggested revisions is to provide a comparison of the dose-response curve to cGMP between mouse and ground squirrel at a negative potential.

We thank the reviewers for prompting us to investigate cGMP dose dependence. We compared dose-response curves, as suggested. The new data are now added as a new Figure 9 in the manuscript, and also mentioned in our response to major point 3 below.

1) CNGA3 is not per se a real cold sensor, rather the channel becomes sensitive to low cGMP levels below a certain temperature threshold. One might argue about the appellation of a cold sensor, as cold acts more like a subthreshold amplifier of the channel function.

We agree with the reviewers. We therefore changed the title to “CNGA3 **acts as a** cold sensor in hypothalamic neurons”, and modified the text accordingly. We believe the new title is independent of the mechanism of cold activation of CNGA3.

2) In Figure 1, authors observed a subpopulation of cold-sensitive neurons that represented 10% and 6% of mouse and ground squirrel total POA neurons population, respectively. The difference, although statistically different, is very minor and suggest that another cold sensor might be expressed in ground squirrel POA neurons. Brain temperature of hibernating ground squirrel can be as low as 5ºC. How do the authors reconcile this with their observation that ground squirrel POA neurons can still respond to cold?

We agree with the reviewers. We have discussed this point as follows:

“Our observation that a small fraction of squirrel POA neurons exhibits cold sensitivity suggests the presence of a CNGA3-independent mechanism, which could be important for active monitoring of body temperature during hibernation (Andrews, 2019).”

3) In Figure 4F authors show that there is no difference for cGMP sensitivity between mouse and ground squirrel CNGA3 in HEK293 cells at +90 mV. However, there seems to be a difference at negative potentials, and in an opposite fashion of what is observed in oocytes.

We measured current values at -90mV, as requested. The new data are added to Figure 7. Overall, the effects of temperature are similar at +90 and -90mV, but the current at -90mV is much smaller due to outward rectification observed in the whole-cell mode.

A comparison of the dose-response curve to cGMP between mouse and ground squirrel at negative potential would be more convincing.

We agree with the reviewers. We performed measurements of cGMP dose response in mouse and squirrel CNGA3 at 22^o^C and 12^o^C in inside-out patches. In summary, mouse and squirrel CNGA3 have the same EC50 for cGMP at 22^o^C, but cold significantly shifts EC_50_ for the mouse channel only from 22 to 4uM. These experiments reveal that cold potentiates mouse CNGA3 by left-shifting EC_50_ for cGMP (new Figure 9). Text as shown below.

“Next, we aimed to clarify the mechanism of cold potentiation of mouse CNGA3, and hypothesized that cold decreases the effective concentration of cGMP required for channel opening. To test this, we investigated the effect of temperature on mCNGA3 activity at different intracellular cGMP concentrations using inside-out patches of HEK293T cells (Figure 9A). Paired recordings from the same patches at different temperatures revealed a significant decrease in halfmaximal cGMP concentration (EC_50_) upon cooling from 21.8±5.48 µM at 22^o^C to 3.5±0.63 µM at 12^o^C (Figure 9C and E). In contrast to mouse CNGA3, and in agreement with our whole-cell data in oocytes and HEK293T cells, we did not detect a significant change in cGMP EC_50_ for squirrel CNGA3 (26.4±1.12 µM at 22^o^C, 20.4±2.14 µM at 12^o^C, Figure 9B, D and E). Interestingly, cooling from 22^o^C to 12^o^C at saturating (≥ 100 µM) cGMP concentrations inhibited maximal activity of mouse CNGA3 by 25%, whereas this effect was significantly higher for the squirrel channel (60%, Figure 9F).[…] Together, these findings reveal that the potentiating effect of cold at low cGMP concentrations is specific to mouse CNGA3, and is caused by a left-shift in EC_50_ for the cyclic nucleotide.”